# Characteristics of Al-Si Alloys with High Melting Point Elements for High Pressure Die Casting

**DOI:** 10.3390/ma13214861

**Published:** 2020-10-29

**Authors:** Tomasz Szymczak, Grzegorz Gumienny, Leszek Klimek, Marcin Goły, Jan Szymszal, Tadeusz Pacyniak

**Affiliations:** 1Department of Materials Engineering and Production Systems, Lodz University of Technology, 90-924 Lodz, Poland; tadeusz.pacyniak@p.lodz.pl; 2Institute of Materials Science and Engineering, Lodz University of Technology, 90-924 Lodz, Poland; leszek.klimek@p.lodz.pl; 3Department of Physical & Powder Metallurgy, AGH University of Science and Technology, 30-059 Krakow, Poland; marcing@agh.edu.pl; 4Department of Technical Sciences and Management, University of Occupational Safety Management in Katowice, 40-007 Katowice, Poland; jszymszal@wszop.edu.pl

**Keywords:** crystallization, microstructure, mechanical properties, thermal and derivative analysis

## Abstract

This paper is devoted to the possibility of increasing the mechanical properties (tensile strength, yield strength, elongation and hardness) of high pressure die casting (HPDC) hypoeutectic Al-Si alloys by high melting point elements: chromium, molybdenum, vanadium and tungsten. EN AC-46000 alloy was used as a base alloy. The paper presents the effect of Cr, Mo, V and W on the crystallization process and the microstructure of HPDC aluminum alloy as well as an alloy from the shell mold. Thermal and derivative analysis was used to study the crystallization process. The possibility of increasing the mechanical properties of HPDC hypoeutectic alloy by addition of high-melting point elements has been demonstrated.

## 1. Introduction

High melting point elements, such as Cr, Mo, V and W are rarely used as additions in Al-Si alloys. Research papers described two main reasons for using these additions in Al-Si alloys. The first reason is the enhancement of the precipitation hardening effect [1,2,3,4] while the second one is a reduction of the detrimental effect of iron on mechanical properties [4,5,6,7,8,9]. However, the use of Cr, Mo, V and W in Al-Si alloys is very limited considering the nature of their interaction with the main alloy constituents, i.e., Al and Si. The phase diagrams Al-Cr [10,11], Al-Mo [12], Al-V [11] and Al-W [11] show the lack of solubility or very limited solubility of high-melting point elements in aluminum. According to [10], Cr is not soluble in Al, while according to [11], the solubility of Cr in Al is negligible and amounts to 0.71 wt % at 661.5 °C. Tungsten is not soluble in Al [11]. Similar behavior shows molybdenum [11,12], while the solubility of vanadium in aluminum is maximum 0.6 wt % (0.3 at %) [11]. Very limited solubility of the above-mentioned elements in aluminum causes the precipitation of a number of intermetallic phases. Also, Si-V [13], Cr-Si [11], Mo-Si [14] and W-Si [15] phase diagrams show that the tested elements practically do not dissolve in silicon. In accordance with these phase diagrams, Cr, Mo, V and W also tend to form numerous intermetallic phases with silicon. For example, four intermetallic phases can occur in the Si-V diagram, i.e., SiV_3_, Si_3_V_5_, Si_5_V_6_ and Si_2_V [13]. On the other hand, the analyzed elements offer excellent mutual solid-state solubility. Cr-Mo, Cr-V, Mo-V, Mo-W and V-W phase diagrams presented in [11,16,17,18,19] show that they form unlimited solutions. In [20], Cr-W phase diagram has been described. It shows that Cr and W form a solid solution (αCr, W) with unlimited mutual solubility of both elements. At 1677 °C, it decomposes into two solid solutions (α1)-rich in chromium and (α2)-rich in tungsten.

The presented data show that in Al-Si alloys containing Cr, Mo, V and W, intermetallic phases with Al and/or Si can crystallize. In multicomponent Al-Si alloys, the possibility of the formation of more complex phases containing other constituents, like Cu, Mg or Ni should be taken into account. Intermetallic phases can significantly increase the brittleness of Al alloys and reduce their strength and elongation. The risk of the precipitation of intermetallic phases in aluminum alloys containing Cr, Mo, V and W increases with the decreasing rate of heat transfer from the casting. Therefore, these additives are not applicable to Al-Si alloys produced in sand and ceramic molds. Additionally, in the case of hypoeutectic alloys, the formation of intermetallic phases in a higher temperature than the temperature of α(Al) phase crystallization changes the additives concentration ahead of the dendritic crystallization front. As a consequence of the high heat transfer during solidification in metal molds, the α(Al) phase can be supersaturated with high-melting point elements. High pressure die casting (HPDC) is a technology widely used in industry and characterized by intensive heat transfer from the casting. Therefore, in relation to hypoeutectic aluminum alloys containing Cr, Mo, V and W, precisely this technology should be seen as an opportunity to effectively increase the mechanical properties of castings made from the above mentioned alloys. The possibility of increasing the properties of HPDC Al-Si alloy by supersaturation of α(Al) phase with high melting point elements is the most innovative aspect of this paper.

The aim of the paper is to confront the phenomena occurring during the crystallization of Al-Si alloy with different content of high-melting point elements with their mechanical properties in the context of developing the probable mechanism of changes in the properties of the alloy. The crystallization process was investigated by thermal and derivative analysis. Optical and scanning microscopy were used to study the alloy microstructure. A point analysis of the chemical composition of selected phases in the microstructure was also performed with the use of an EDS (Energy Dispersive Spectroscopy) detector. The static tensile test and the Brinell hardness test were used to determine the mechanical properties

## 2. Materials and Methods 

A typical hypoeutectic Al-Si alloy for high pressure die casting, i.e., EN-AC 46000 (EN-AC AlSi9Cu3 (Fe)), was used for the tests. It is included in the PN-EN 1706 standard. The chemical composition of the base alloy is shown in Table 1. The range of the chemical composition is derived from 15 specimens of the base alloy. The reason for such a number of specimens is the testing of 15 different combinations of high melting point elements. The number of Cr, Mo, V and W combinations means all statistically possible connections that can be used for the four analyzed elements. Therefore, each time new base alloy was smelted from ingots of EN AC-46000 alloy. The individual connections Cr, Mo, V and W have been shown in the following description of a melting technology.

The base alloy was melted in a gas-heated shaft furnace with a maximum charge capacity of 1.5 tonnes (StrikoWestofen, Gummersbach, Germany). After smelting, it was refined inside the shaft furnace. An Ecosal Al 113S solid refiner was used for this treatment. After tapping the alloy from the furnace into a ladle, it was deslagged with an Ecremal N44 deslagging agent. After deslagging, the liquid alloy was transferred to a holding furnace set up next to an Idra 700S horizontal cold chamber pressure die casting machine (Idra, Travagliato, Italy). In the holding furnace, AlCr15, AlMo8, AlV10 and AlW8 master alloys were added into the alloy. The temperature in the holding furnace was 750 °C. After dissolution of the master alloys, the temperature of the liquid metal was reduced to a value appropriate for the test casting process. High melting point elements, i.e., chromium, molybdenum, vanadium and tungsten, were added as single elements, in double combinations (CrMo, CrV, CrW, MoV, MoW and VW), in triple combinations (CrMoV, CrMoW, CrVW and MoVW), or all the additives were added simultaneously. For individually added Cr, Mo, V and W, their content was kept in the range of 0.0–0.5 wt %, and it was increased in 0.1 wt % steps. If more than one high melting point element was added, all the elements were used in equal amount. For double combinations, the content of additives was in the range of 0.0–0.4 wt %, and in subsequent melts it was increased in 0.1 wt % steps. In the case of triple and quadruple combinations, the content of elements was in the range of 0.00–0.25 wt %, and in subsequent melts it was increased in 0.05 wt % steps. From the resultant alloys, covers of roller shutter housings with a predominantly 2 mm wall thickness were made by the high pressure die casting process.

For each chemical composition tested, three specimens were taken from one high pressure die casting to carry out the tensile test. The specimens had a flat shape and 2 × 10 mm^2^ rectangular cross-section. This cross-section is recommended by PN-EN 1706 standard for testing the strength of high pressure die casting. An Instron 3382 machine (Instron, Norwood, MA, USA) was used to perform the tensile test at a speed of 1 mm/min. The test enabled the determination of the tensile strength R_m_, the yield strength R_p0.2_ and the elongation A. Alloy hardness was measured by the Brinell method using an HPO-2400 hardness tester (WPM LEIPZIG, Leipzig, Germany) according to PN EN ISO 6506. The diameter of the ball was 2.5 mm, the load was 613 N, and the static load holding time was 30 s.

Thermal and derivative analysis was used as a tool to study the solidification process. Thermal and derivative curves were recorded with a PtRh10-Pt thermocouple enclosed by a quartz tube placed in the thermal center of a probe made of a resin coated sand. Its dimensions are shown in Figure 1. The alloy was superheated to 1000 °C before the probe was filled with liquid alloy.

Metallographic specimens for microstructure examinations were prepared on specimens taken from castings made in the resin coated sand probe and by high pressure die casting. The surface of the specimens was etched with a 2% HF acid solution. The microstructure was examined using Nikon Eclipse MA200 optical microscope (Nikon, Tokyo, Japan) with ×100 and ×1000 magnification for castings made in the resin coated sand probe and by high pressure die casting process, respectively. The applied magnification ensures correct visibility of all alloy phases in the examined area.

Point microanalysis of the element concentration was performed using the Pioneer EDS detector cooperating with the HITACHI S-3000N scanning electron microscope (Hitachi, Tokyo, Japan) and the VENTAGE software from NORAN. The specimens for scanning under the scanning microscope were cut from the test castings using a hand hacksaw and water cooling. The shell mold casting specimens had the shape of a cube with a side of 10 mm. Specimens taken from high pressure die castings had the shape of a cuboid with sides 2/10/20 mm, and after cutting they were mounted in a conductive phenolic resin. After mounting, the specimen was cylindrical with a diameter of 36 mm and a height of 12 mm. The specimens were then ground using sandpaper and polished using diamond suspensions with a gradation of 9 to 1 µm.

The area of the spot examination of the elements concentration is shown in the figures of the microstructure made with the use of a scanning microscope. These images show the intermetallic phases within which the analyzes were performed. For the purpose of this study, phase precipitations of relatively large sizes were selected in order to avoid the analytical beam going beyond the area of the tested phase.

## 3. Results and Discussion

Figure 2 compares the thermal and derivative analysis curves of the EN AC-46000 base alloy and alloy containing 0.5 wt % Mo and Cr, Mo, V and W added in an amount of 0.25 wt % each.

Studies have revealed a three-stage solidification process of the base EN EN-46000 alloy from the resin coated sand probe. The first stage taking place at the highest temperature is the crystallization of α(Al) dendrites solid solution. This process is marked by the thermal effect C_s_AB. The primary crystallization of α(Al) dendrites directly from the liquid results from Al-Si phase diagram shown in Figure 3. 

The presented Al-Si phase diagram shows that after the crystallization of α(Al) dendrites, the crystallization of α(Al) + β(Si) eutectic mixture takes place. The tested AlSi9Cu3(Fe) alloy is multicomponent, therefore more complex eutectic mixtures crystallize in it. In these mixtures, apart from α(Al) and β(Si) phases, there are also intermetallic phases. The crystallization of multicomponent Al-Si alloys with this type of complex eutectic mixtures was described in detail in [22,23,24]. In accordance with the knowledge contained therein, the authors present the further crystallization process of the analyzed alloy.

When dendrite crystallization is complete, the complex α(Al) + Al_15_(Fe,Mn)_3_Si_2_ + β(Si) (ternary) and α(Al) + Al_2_Cu + AlSiCuFeMnMgNi + β(Si) (quaternary) eutectic mixtures crystallize. Thermal effects caused by the crystallization of the above mentioned eutectic mixtures have been marked with the symbols BEH and HKL, respectively. The designation of AlSiCuFeMnMgNi phase should be treated as conventional, because it does not mean an intermetallic phase containing all the elements included, but a number of phases that may contain these elements. The chemical composition of this phase depends on the diversified (as a result of microsegregation) content of elements in different areas of the residual liquid. It can be Mg_2_Si phase or the phases from Al-Cu-Ni, Al-Fe-Mn-Si and Al-Cu-Mg-Si systems. The microstructure of the base alloy and containing Cr, Mo, V and W is presented in detail in [25]. This paper presents only those aspects of microstructure formation that are related to the mechanism of its strengthening presented by the authors.

When the additives in an amount not exceeding 0.1 wt.% Cr, 0.4 wt % Mo or W, and 0.5 wt % V are added as single elements into the base alloy, there are no new thermal effects on the thermal and derivative curves and new phases are not formed in the microstructure. In this range of the content, the above mentioned elements tend to join the phases already existing in the base alloy. These are mainly iron-rich intermetallic phases occurring in both eutectic mixtures, designated as Al_15_(Fe,Mn,M)_3_Si_2_, where M is any high melting point element or any combination of them. A single addition of chromium in an amount of 0.2 wt % and molybdenum or tungsten in an amount of 0.5% leads to primary crystallization of the Al_15_(Fe,Mn,M)_3_Si_2_ phase. Vanadium in the tested content range did not cause this phase to crystallize. As a result of the crystallization of the primary Al_15_(Fe,Mn,M)_3_Si_2_ phase on the derivative curve a thermal effect is created, which does not have a distinct local maximum. An example of such an effect is visible on the derivative curve of an alloy containing 0.5 wt % Mo (Figure 2–red color) and is described as C_s_A’. Increasing the amount of high melting point elements produces a thermal effect with a clearly visible local maximum. This thermal effect is shown in Figure 2 (blue color), for an alloy containing 0.25 wt % Cr, Mo, V and W. It is described as C_s_A’A”. The amount and size of this phase increases with the increase in the content of high melting point elements in the alloy. Figure 4 shows the microstructure of the base alloy and containing 0.25 wt % Mo, V and W each with marked component phases.

In Figure 4b Al_15_(Fe,Mn,M)_3_Si_2_ phase is characterized by a skeletal morphology. For example, Figure 5 shows the chemical composition of the analogous phase in the alloy containing 0.4 wt % Cr.

There is a high concentration of Al (57.7 wt %) and an increased concentration of Fe (17.7 wt %) in the phase; Si (10.8 wt %); Cr (7.53 wt%) and Mn (5.9 wt %). Its phase chemical composition indicates that it is Al_15_(Fe,Mn,Cr)_3_Si_2_ phase. The occurrence of this phase in alloys containing Cr has been confirmed in [26,27,28]. The chemical analysis of Al_15_(Fe,Mn,M)_3_Si_2_ phase detected in alloys containing additionally Mo, V and W showed the presence of all these elements. Thus, the crystallization of these intermetallics causes the depletion of the remaining liquid in high melting point elements. A relatively large amount of the primary Al_15_(Fe,Mn,M)_3_Si_2_ phase causes the depletion of the liquid in high melting point elements leading to crystallization the classic plate α(Al) + β(Si) eutectic mixture instead of the ternary one. Thus, high melting point elements first increase their concentration ahead of the crystallization front of α(Al) phase, but then reduces it as a result of the primary crystallization of Al_15_(Fe,Mn,M)_3_Si_2_ phase. The possibility of supersaturation of α(Al) dendrites is related to the concentration of the dissolved substance ahead of the dendrite crystallization front [29] and the rate of heat transfer during the solidification process. In [30,31], general relationships (1) and (2) were developed to link the crystal nucleation rate and growth rate, respectively, with the supersaturation of the crystal with the substance on the crystallization front: B = k_b_ × Ω^b^,(1)
V = k_v_ × Ω^v^,(2)
where:B-crystal nucleation rate,V-crystal growth rate,Ω-supersaturation,k_b_ and k_v_-original constant rate of crystal nucleation and growth, respectively,and v-supersaturation exponent for crystal nucleation and growth, respectively.

From the above mentioned relationships it follows that an increase in the crystal nucleation rate and growth rate increases the degree of supersaturation. It is also generally known that that the higher rate of heat transfer leads to the higher rate of the nucleation and growth of the crystal. Therefore, two factors can accelerate supersaturation–a high concentration of high melting point elements on the dendritic crystallization front and the high rate of heat transfer during crystallization. Based on 84 thermal and derivative tests for the base alloy as well as all variants of the high melting point elements used, the average heat transfer rate during the crystallization of α(Al) phase was calculated. The value obtained was 0.13 °C/s. This, relatively low, heat transfer rate in castings from shell molds creates rather modest possibilities of α(Al) supersaturation with high melting point elements. However, the extensive thermal and derivative tests showed an increase in the crystallization start temperature C_s_ as a result of addition high melting point elements and increasing their content in the alloy. This was the case for all analyzed combinations of these elements. Figure 6 shows the effect of high-melting elements on the crystallization start temperature ΔtCs, for individually and simultaneously added Cr, Mo, V and W.

Figure 6a shows that all individually added high melting point elements increase the liquidus temperature. This is mainly due to the pre-dendritic crystallization of Al_15_(Fe,Mn,M)_3_Si_2_ phase, which occurs when 0.2 wt % Cr and 0 5 wt.% Mo and W are added (thermal effects C_s_A’ or C_s_A’A”). Crystallization of this phase is responsible for a rapid increase in the temperature ΔtC_s_. However, for each high melting point element analyzed, the increase in C_s_ temperature was also obtained in the range of their concentration that did not cause pre-dendritic crystallization of Al_15_(Fe,Mn,M)_3_Si_2_ phase. This tendency is clearly visible when V, Mo and W are added. Vanadium did not cause pre-dendritic crystallization of Al_15_(Fe,Mn,M)_3_Si_2_ phase, while Mo and W did not cause this phase to crystallize to 0.4 wt %. In the above-mentioned range of high melting point elements, the increase of ΔtC_s_ is relatively mild. When Cr, Mo, V and W are simultaneously added (Figure 6b), up to 0.05 wt % each, the increase in ΔtCs is also initially mild, but starting with the 0.10 wt %, a rapid jump and faster growth occur. This rapid increase in ΔtC_s_ is due to the pre-dendritic crystallization of the Al_15_(Fe,Mn,M)_3_Si_2_ phase. The increase in the liquidus temperature at the content of high melting point elements insufficient to start the primary crystallization of Al_15_(Fe,Mn,M)_3_Si_2_ phase indicates transfer of a certain amount of these elements to α(Al) dendrites. Much better conditions for supersaturation of the α(Al) solid solution with high melting point elements are provided by the high pressure die casting (HPDC) technology, where the rate of heat transfer from the casting is much higher. The rate of heat transfer during crystallization of the α(Al) dendrites in HPDC alloy was determined from relationship (3) presented in [32]:SDAS = 39.4 × V^(−0.317),(3)
where:SDAS-secondary dendrite arms spacing in α(Al) phase, µm;V–the rate of heat transfer during the crystallization of α(Al) dendrites, °C/s.

The measured average SDAS in the examined HPDC alloy was 10.65 µm. The rate of heat transfer calculated for SDAS from relationship (3) amounts to ~55 °C/s. This value is over 400 times higher than the value obtained for alloy from shell mold. The high rate of heat transfer from the HPDC alloy produces more refined microstructure compared with the microstructure obtained in shell mold. Figure 7 shows the microstructure of HPDC EN AC-46000 alloy containing Cr, V and W, each in an amount of 0.25 wt %. The constituent phases are also marked.

The microstructure of the die cast base alloy consists of α(Al) phase dendrites and eutectic mixture. The eutectic mixture consists of α(Al) and β(Si) solid solutions and relatively fine intermetallic phases. Diffraction tests have shown the presence of intermetallic phases: Al_2_Cu, Al_2_CuMg and other phases that crystallize in Al-Fe-Si, Al-Cu-Fe and Mn-Ni-Si phase systems. The addition of a certain amount of high melting point elements causes the pre-dendritic crystallization of the Al_15_(Fe,Mn,M)_3_Si_2_ phase. With the increasing content of above mentioned elements, the amount of this phase also increases. Figure 7b shows this phase assume the form of polygons or stars. The size of the precipitates, depending on the content of high melting point elements, is from a few to ~50 µm. Analysis of the chemical composition of the pre-dendritic Al_15_(Fe,Mn,M)_3_Si_2_ phase has shown that it can “assimilate” all high melting point elements tested. Figure 8 presents the results of EDS point analysis carried out within the area of the pre-dendritic phase in the alloy containing Cr, Mo, V and W in an amount of 0.25 wt.% each.

The EDS analysis shows that the tested phase is most effective in assimilating Cr (6.44 wt %), slightly less effective in Mo (3.72 wt %) and V (2.96 wt %), and the least effective in W (0.99 wt %). The total concentration of Cr, Mo, V and W in this phase is ~14 wt %. The “assimilation” of chromium, molybdenum, vanadium and tungsten by the Al_15_(Fe,Mn,M)_3_Si_2_ phase causes the depletion of these elements in the liquid. Since the Al_15_(Fe,Mn,M)_3_Si_2_ phase crystallizes directly in front of the α(Al) dendrites, in the liquid ahead of the dendritic crystallization front, similar changes occur in the concentration of Cr, Mo, V and W as in the alloy from the shell mold. Chromium, molybdenum, vanadium and tungsten in the range of concentration that did not cause the pre-dendritic crystallization of Al_15_(Fe,Mn,M)_3_Si_2_ phase leads to an increase in the concentration of these elements ahead of the crystallization front of α(Al) dendrites. Further increase in the content of above mentioned elements to a level that can trigger the crystallization of the pre-dendritic Al_15_(Fe,Mn,M)_3_Si_2_ phase reduces the Cr, Mo, V and W concentration ahead of the crystallization front of α(Al) dendrites. The described changes in the concentration of Cr, Mo, V and W ahead of the crystallization front of α(Al) dendrites can significantly affect the mechanical properties of HPDC alloy. The relatively high cooling rate during the crystallization of α(Al) dendrites, (~55 °C/s), leads to supersaturation of the α(Al) phase with chromium, molybdenum, vanadium and tungsten, and may increase the mechanical properties of HPDC alloy. This is confirmed by the results of the statistical analysis of the effect of Cr, Mo, V and W on the tensile strength R_m_; yield strength R_p0.2_; elongation A and Brinell hardness of HPDC alloy presented in [33,34,35]. Statistical analysis was performed at the significance level p (α) = 0.05. The analysis of variance (ANOVA) test for the main effects was used to assess the influence of high melting point elements on the mechanical properties. The values of strength properties in [33,34,35] were presented in the standardized and dimensionless form. The results of the statistical analysis showed that each high melting point element can cause an increase in R_m_, R_p0.2_; A and HB. The data presented in these papers shows an unambiguously analogous nature of the impact of the tested high melting point elements on the properties of this alloy. For the purposes of this paper, standardized quantities were decoded to the form of real values, which can be expressed in the units of measurement proper for the tested mechanical properties. Figure 9 presents an example of the effect of high melting point elements on the R_m_, R_p0.2_; A and HB. The error bars shown in the graphs represent the values of the Standard Error of the Mean (SEM). The small SEM values shown in (Figure 9) prove the correctness of the results, which indicate the effect of high melting point elements on the tested properties.

Figure 9a shows that Cr at the level of 0.05 wt % allows to obtain the tensile strength R_m_ = 274.4 MPa. It is a value by ~13% higher in relation to the base alloy. Other high melting point elements make it possible to obtain a high R_m_ value with a content 0.05–0.10 wt %. 0.05 wt % vanadium has the greatest effect on the increase in R_m_. This elements results in R_m_ = 277 MPa, which means an increase by ~14% relative to the base alloy. 

The highest yield strength is obtained with the addition of high melting point element in an amount of 0.05 wt %. The element most effective in increasing the value of R_p0.2_ is vanadium (Figure 9b). Its presence produces R_p0.2_ = 126.8 MPa, which means an increase by ~12% relative to the base alloy. The highest elongation is usually obtained when the amount of high melting point element is 0.15 wt %. The greatest impact on the increase in elongation has 0.15 wt % Mo. The elongation is then 5.30%, which gives a 40% increase relative to the base alloy. The data presented in Figure 9c shows that chromium in an amount of 0.15 wt % is also very effective in increasing the elongation to A = 5.21%. Relative to the base alloy, it is a 36% increase.

To obtain high values of the Brinell hardness, the optimal content of high melting point elements is 0.05 wt % for chromium, molybdenum and vanadium, and 0.5 wt % for tungsten. The impact of V and W on HB levels is shown in Figure 9d. The greatest impact on HB increase has 0.5 wt.% W (117 HB). The relative increase is rather small and amounts to 5%.

The highest values of R_m_ and R_p0.2_ are the result of a relatively high supersaturation of α(Al) solid solution with these elements. The effect of this phenomenon on the strength properties is most beneficial, when the high melting point elements are added in an amount insufficient to start the primary crystallization of Al_15_(Fe,Mn,M)_3_Si_2_ phase. The supersaturation of α(Al) dendrites is also beneficial for the hardness of alloy containing 0.05 wt % Cr, Mo or V. The highest hardness obtained at 0.5 wt % W was due to the presence of relatively large precipitates of Al_15_(Fe,Mn,W)_3_Si_2_ phase. The largest increase in elongation accompanying the addition of high melting point elements in an amount of 0.15 wt % was the result of primary crystallization of relatively small and compact precipitates of Al_15_(Fe,Mn,M)_3_Si_2_ phase, which reduced the concentration of high melting point elements as well as Fe and Mn ahead of the crystallization front of α(Al) dendrites. Crystallization of Al_15_(Fe,Mn,M)_3_Si_2_ phase in alloy with high melting point elements can reduce the supersaturation of α(Al) dendrites compared with the alloy without them. This leads to an increase in the alloy plasticity.

Statistical analysis in [33,34,35] gives no evidence supporting the occurrence of a synergistic effect between simultaneously introduced high melting point elements and alloy properties. However, the results of mechanical tests carried out on various combinations of the co-introduced high melting point elements indicate the possibility of obtaining much higher values of R_m_, R_p0,2_ and A than the values obtained in a statistical analysis carried out for the same additives but introduced as single elements. The largest increase in R_m_ was obtained with the addition of vanadium and tungsten added in an amount of approximately 0.2 wt % each. The value of R_m_ was then 299 MPa, which means that it was higher by ~50% relative to the base alloy. The same addition of V and W also caused the largest (over twofold) increase in an elongation. The maximum increase in R_p0.2_, i.e., by ~21%, was due to the use of Cr and W added in an amount of 0.1 wt.% each.

## 4. Conclusions

From the data presented in this paper, the following conclusions emerge:Proper amount of high melting point elements added into HPDC hypoeutectic alloy as well as alloy from shell molds results in pre-dendritic crystallization of the Al_15_(Fe,Mn,M)_3_Si_2_ phase.High intensity of the heat transfer during HPDC process enables supersaturation of α(Al) dendrites with high melting point elements.The addition of Cr, Mo, V and W as well as pre-dendritic crystallization of the Al_15_(Fe,Mn,M)_3_Si_2_ phase change concentration of the above mentioned elements ahead of the crystallization front of α(Al) dendrites, resulting in its various supersaturation during HPDC process.A significant increase in mechanical properties of the tested hypoeutectic alloys was obtained by supersaturation of the α(Al) phase with high melting point elements during HPDC process.Changes in mechanical properties of the tested alloys were largely affected by supersaturation of the α(Al) phase, and also by the size of precipitates and content of the pre-dendritic Al_15_(Fe,Mn,M)_3_Si_2_ phase.In the future, to prove the correctness of the presented hypothetical mechanism of strengthening of HPDC alloy by supersaturation of α(Al) phase with high melting point elements, analyzes should be carried out to show the presence of these elements in α(Al) phase with their different contents.

## Figures and Tables

**Figure 1 materials-13-04861-f001:**
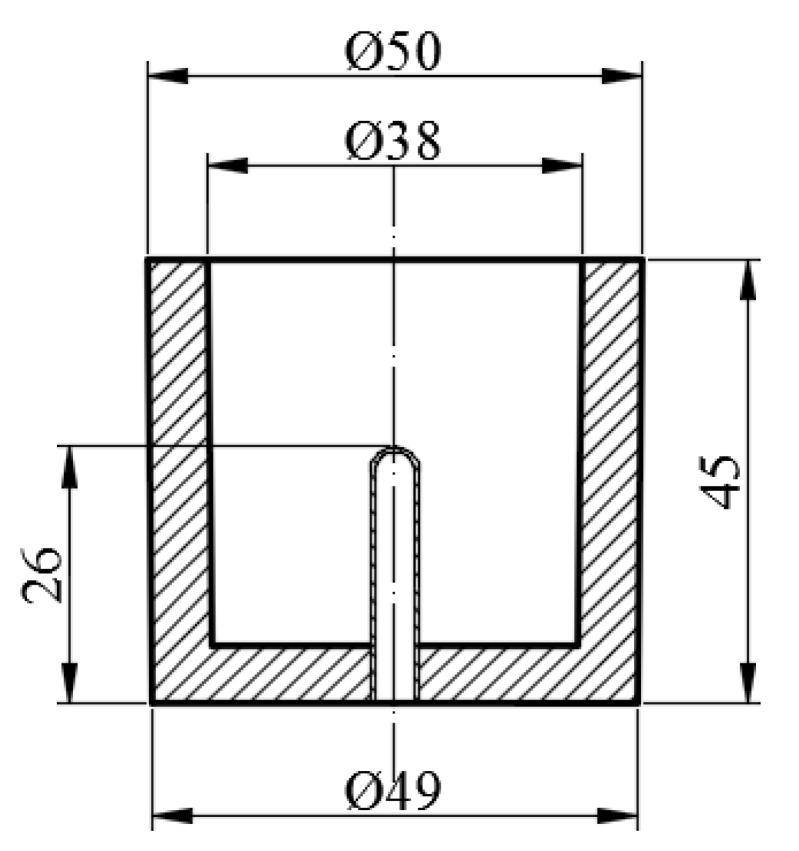
Dimensions of the resin coated sand probe (units: mm).

**Figure 2 materials-13-04861-f002:**
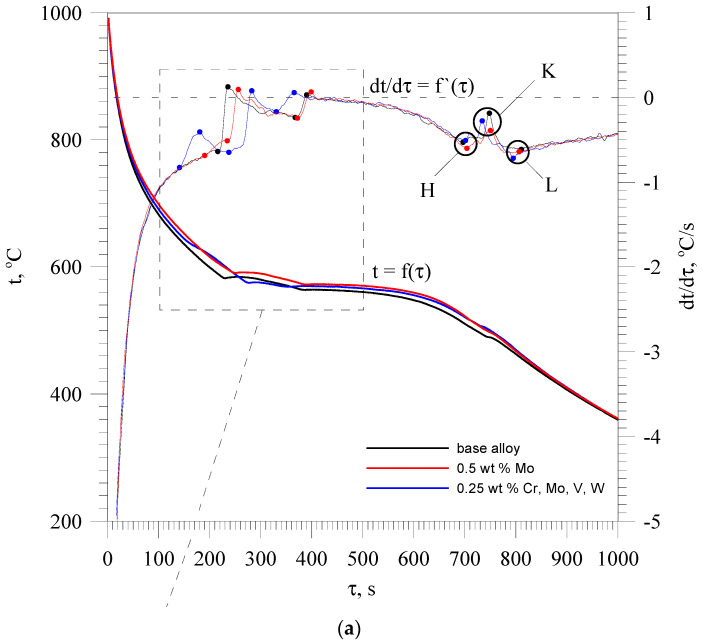
Thermal and derivative curves of the base alloy (black), alloy containing 0.5 wt % Mo (red) and alloy containing Cr, Mo, V and W in an amount of 0.25 wt % each (blue). (**a**) and their enlarged fragment (**b**).

**Figure 3 materials-13-04861-f003:**
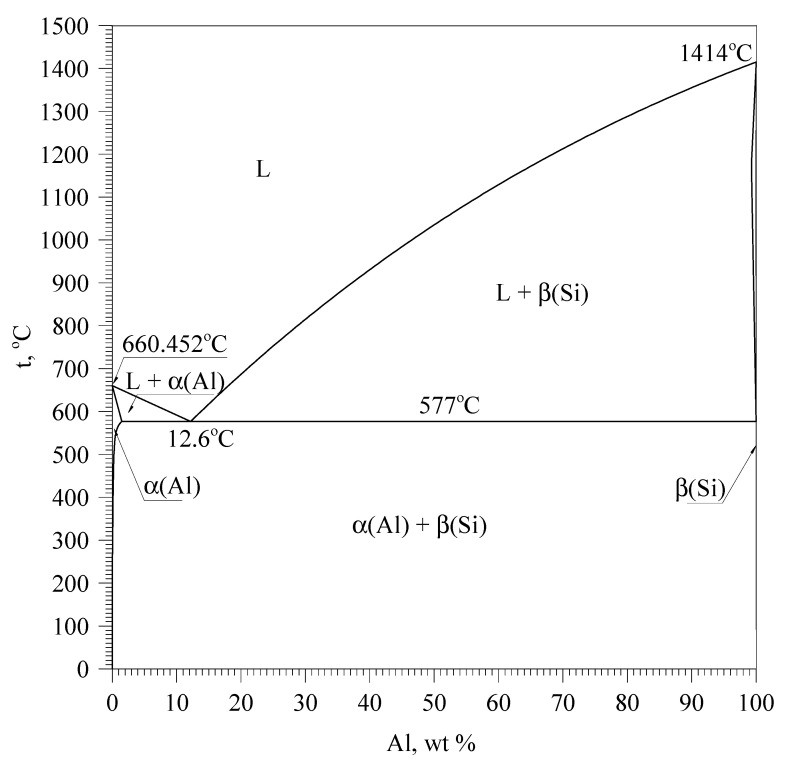
Al-Si phase diagram (data from [11,21,22]).

**Figure 4 materials-13-04861-f004:**
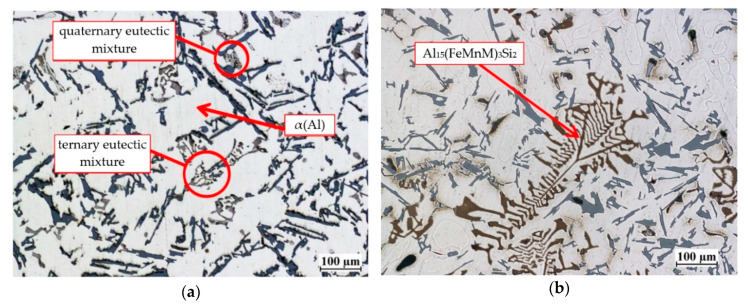
The microstructure of: (**a**) EN AC-AlSi9Cu3(Fe) base alloy, (**b**) alloy containing Mo, V and W in an amount of 0.25 wt. % each.

**Figure 5 materials-13-04861-f005:**
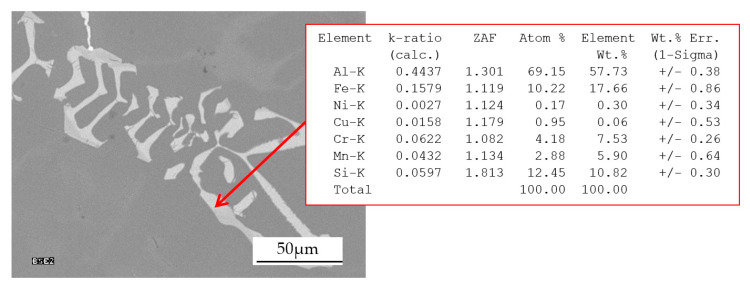
An example of the area with Al_15_(Fe,Mn,M)_3_Si_2_ phase in the alloy containing 0.4 wt % Cr from the shell mold and the results of the point measurement of the chemical composition within it.

**Figure 6 materials-13-04861-f006:**
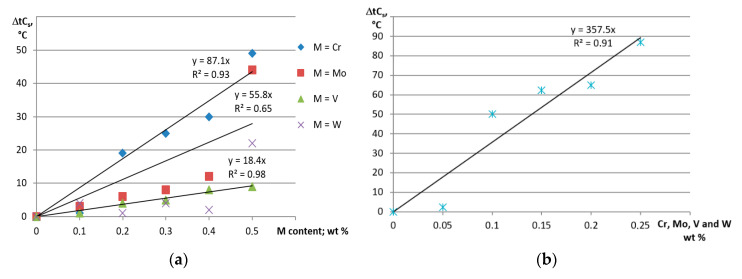
Changes in the crystallization temperature ΔtC_s_ caused by the variable content of Cr, Mo, V or W added: (**a**) as single elements and (**b**) jointly; M–any high melting point element tested, i.e., Cr, Mo, V or W.

**Figure 7 materials-13-04861-f007:**
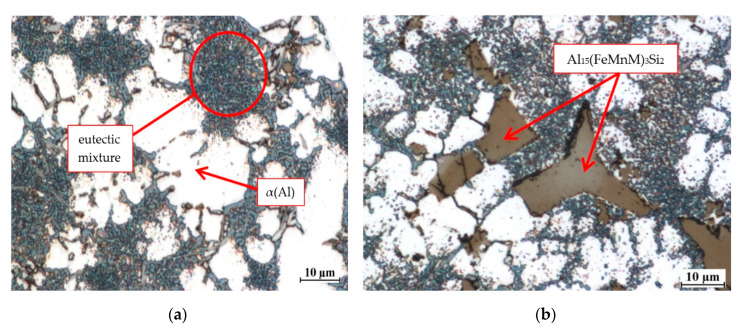
The microstructure of: (**a**) base alloy and (**b**) alloy containing Cr, V and W in an amount of 0.25 wt % each.

**Figure 8 materials-13-04861-f008:**
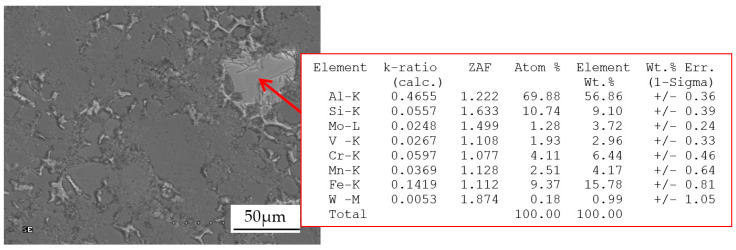
An example of Al_15_(FeMnM)_3_Si_2_ phase in HPDC alloy containing Cr, Mo, V and W in an amount of 0.25 wt % each and the results of a point analysis of the chemical composition within it.

**Figure 9 materials-13-04861-f009:**
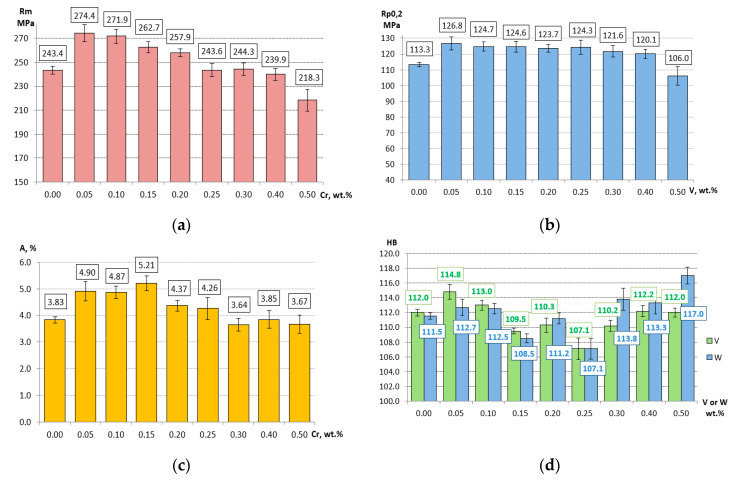
Examples of the effect of selected high melting point elements on the mechanical properties: (**a**) the effect of Cr on R_m_ level; (**b**) the effect of V on R_p0.2_ level; (**c**) the effect of Cr on A level, (**d**) the effect of V and W on HB level.

**Table 1 materials-13-04861-t001:** Chemical composition of the base EN AC-46000 alloy.

Chemical Composition, wt %									
**Si**	**Cu**	**Zn**	**Fe**	**Mg**	**Mn**	**Ni**	**Ti**	**Cr**	**Al**
8.69–9.35	2.09–2.43	0.90–1.07	0.82–0.97	0.21–0.32	0.18–0.25	0.05–0.13	0.042–0.049	0.023–0.031	rest

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
