# Peer review of "Characteristics of Al-Si Alloys with High Melting Point Elements for High Pressure Die Casting"

_materials, 2020, doi:10.3390/ma13214861_

Round 1

Reviewer 1 Report

This manuscript studied the influence of high-melting point elements on mechanical properties to Al-Si alloys using high pressure die casting (HPDC). However current manuscript needs much improved before considering possible publication in Materials. More details are listed below:

1. There is a lack of novelty. The separate addition of these elements to Al-Si has been studied for a long time. The effects of combined additions have less descriptive. The author shall explain the innovation of this paper.

2. Mechanical properties descriptions are confusing. In Fig. 8, the horizontal coordinates of the four graphs do not match. The comparison between the effect of Cr element to tensile strength and the effect of V element to yield strength is not adequate. The authors must give the full mechanical properties or give evidence that the elements are interchangeable.

3. There are several proprietary symbols and grammatical errors in the text. The writing shall be much improved with the aid by native speaker or language editing services for possible publication. For example,

Line 110 on page 3: “The alloy was superheated to 1000°C before the probe was filled with liquid alloy.” A single sentence is not normally treated as a paragraph.

Fig. 2 on page 4: The vertical coordinate “t, ° C”. The temperature shall use the symbol “T” rather than “t”.

Line 282 on page 9: “This elements results in Rm = 277 MPa”, and this shall be revised as “This element results in Rm = 277 MPa”.

Reviewer 2 Report

Title: Characteristic of Al-Si alloys….

Manuscript ID: materials-977421

Authors: Szymczak et al.

Dear Authors,

Thank you for the opportunity to read your article. I found it has an interesting potential, and there are some results presented in order to prove your approach is valid. On the other hand, materials and methods need more clarification while fair and scientific discussion should be made using your results that are not fully utilized. I suggest this article be revised throughout before resubmission for another review process. As a conclusion, I recommend its major revision at this state.

I hope my comments are helpful.

Good luck,

A reviewer

Major concerns:

“Title”

“Characteristic of Al-Si alloys….casting (HPDC)”-> Characteristics of Al-Si alloys….casting

You studied different characteristics. No need to add abbreviation in the title (but you can do it in introduction).

 “Keywords”

-Please replace keywords with the ones not used in the article title.

 “Introduction”

-Please briefly introduce your methodologies, in relation to your objectives.

“2. Materials and Methods”

-Lines 71-72: “…the testing of 15 different combinations...”->Please provide the list of 15 combinations, and also provide the justifications of those selections.

- Table 1: The way numbers are given there is confusing. For example, Si wt.%, do you mean it is in between 8.69 and 9.35 wt.%? If so, I would suggest you write 8.69-9.35 instead. For Al, what does “al.” mean?

-Line 76: “…a maximum charge…”->…the maximum charge…

-Lines 117-119: Please write about your specimen (e.g. dimension) fed into the SEM chamber, specific sample preparation (e.g. cutting, polishing, coating), and EDS measurement conditions (e.g. accelerating voltage, beam current intensity, working distance, acquisition time) if any. You may read the following articles.

https://doi.org/10.1016/j.actamat.2005.12.014

https://doi.org/10.3390/electronics8101202

“3. Results and Discussion”

-Lines 130-131: “When dendrite crystallization is complete…eutectic mixtures crystallize.”->It would be recommended to provide a phase diagram to explain and support your statement.

-Fig.7 (and elsewhere): How representative this image is? The large grain you analyzed is very minor in the image you show.

-Fig.8: Can you have error bars in this figure and discuss the effects of variables studied on discussed values? For example, line 280 stated “13% higher in relation to the base alloy” and line 283 stated “5% relative to the base alloy”, but are they outside of error range or not?

“Conclusions”

-You may inform the possible future work.

Minor concerns:

-English needs to be further polished. You may also use my comments for this purpose.

Round 2

Reviewer 2 Report

Dear Authors,

Now, I see all the suggestions/concerns were addressed, and suggest the journal accept the article for its publication.

Best regards,

A reviewer